# Comparative Cost Functions Analysis in the Construction of a Reference Angular Motion Implemented by Magnetorquers

**Anna Okhitina \*, Stepan Tkachev**  **and Dmitry Roldugin** 

Keldysh Institute of Applied Mathematics RAS, Miusskaya Sq., 4, Moscow 125047, Russia;
stevens_l@mail.ru (S.T.); rolduginds@gmail.com (D.R.)
\* Correspondence: anna.ohitina@mail.ru

**Abstract:** This paper considers a construction procedure of a satellite reference angular motion in the vicinity of an unstable gravitational equilibrium position. The satellite is stabilized on the reference trajectory by the magnetic coils. The problem is solved in several stages. An optimization problem to be solved by the particle swarm optimization method is formulated at each stage. Cost functions are based on the linearized model analysis. The main stage is the construction of a special reference motion, which provides the minimum control torque projection on the geomagnetic induction vector. Optimal geomagnetic field dipole approximation for a given time interval is considered to reduce tracking errors. The paper compares combinations of different cost functions in terms of the terminal attitude accuracy in the presence of perturbations.

**Keywords:** attitude control; magnetic control; three-axis stabilization; periodic motion; disturbance; particle swarm optimization; Floquet theory

---

## 1. Introduction

The ability to formulate and formalize correctly is almost half of the solution of any problem. Formalization of the problem—that is, the translation from the "human" language into the language of mathematics—can be proposed in several ways. For example, in the construction and tracking of trajectories for robots, minimization problems utilize cost functions that describe the modulus of the position difference, integral difference characteristic, root-mean-square deviation, or more complicated ones, as in [1–5]. These cost functions characterize the position error in different ways. It is important to establish which cost function in each specific task provides the best formalization, the best result, and why.

This paper utilizes an active magnetic attitude control system (MACS) to provide three-axis spacecraft (SC) stabilization. This problem was proposed in [6,7] and is widely considered today. The MACS has low power consumption and is easy to manufacture. It is well suited to reduce the cost of missions and makes SC hardware simpler and more reliable [8,9]. This is necessary due to the growth in the number of space-related scientific and applied problems in various industries, as well as educational and new technology demonstration projects. However, the MACS has a significant drawback: It is impossible to realize the component of the required control torque directed along the geomagnetic induction vector **B**. Many works combine the use of the magnetic control and other actuators and concepts to bypass this limitation. For example, [10,11] proposed a method for spacecraft attitude stabilization that simultaneously uses a magnetic attitude control system and the electrodynamic effect of the influence of the Lorentz forces.

However, as the satellite moves along the orbit, the vector **B**'s direction changes, so in general the system is controllable [12,13]. The Lyapunov feedback law is usually used when constructing a control using only an MACS. The main problem here is the correct choice of the control gains. The papers [14–18] offer different solutions. However, even the

most accurate selection of the control gains does not allow a simple feedback law to provide effective suppression of local disturbances. This results in a low stabilization accuracy, which is expected and flight-proven [19] to be around 10–20 degrees.

In some papers, certain modes of motion that are fully controlled by the MACS are considered, and then the corresponding control law is constructed and its performance is proven. In [20] the possibility of constructing an oscillatory control is studied so that a satellite with only an MACS can provide complete control along three axes, regardless of the position of the satellite. For this purpose, Lie-bracket-based controls are being considered. In [21], a proof of global exponential stability is obtained for the magnetic control, which stabilizes the satellite in the desired rotation around the main axis of inertia. The paper [22] presents a spin-stabilization algorithm for an axisymmetric spacecraft using only an MACS. It is shown that one magnetorquer that is perpendicular to the spin axis is enough to stabilize the satellite in the inertial space, and that the satellite's motion remains stable even with control outages. The paper [23] proposes a modification of the B-dot magnetic control strategy that allows the satellite to control the rotation rate. A Lyapunov function is used to prove the asymptotic stability in the spin acquisition phase. Analytical exact solutions of differential equations of the dual-spin spacecraft angular motion under the action of the magnetic restoring torque are obtained in [24].

The paper [25] proposes a method to construct a special reference angular trajectory by minimizing the root-mean-square deviation of the control torque projection onto the geomagnetic induction vector. When the projection is equal to zero, the trajectory is completely controllable, which minimizes the reference angular motion tracking errors. The construction of the trajectory in [25] is performed in a simplified model; in particular, there are no disturbances, and the direct dipole model is used to describe the geomagnetic field. This adversely affects the attitude accuracy in the full model.

The paper [26] describes an alternative approach of a reference trajectory construction. The quadratic programming problem is formulated for the trajectory coefficients under the condition that the projection of the control torque on the geomagnetic induction vector is equal to zero. The constraint is linearized assuming that the trajectory angles are small. In the case of linear systems, the resulting trajectory is the best, since it is completely controllable. However, its application for the nonlinear model provides worse results.

This paper proposes a number of alternative cost functions based on the analysis of a linearized model of the angular motion. In [25], the cost functions were chosen empirically. The innovation of this article is a justification for the choice of cost functions using the Floquet theory. According to this procedure, new cost functions were derived, which resulted in better and more reliable results. In addition, the problem of optimal approximation of the geomagnetic field model for a given time interval is posed. These problems, similarly to [25], are solved using the particle swarm optimization method (PSO) [27–29]. This method is based on a decision-making model of particle motion in a search for the best solution to the optimization problem. Each particle computes the cost function value for its position in the search space of the problem parameters and receives the information about possible best solutions from its neighbors. This method effectively handles cost functions that cannot be represented explicitly and, therefore, cannot be optimized with gradient methods.

An extensive numerical simulation is carried out with various external disturbances in the full model, taking into account the inaccuracy of knowledge of the spacecraft's inertia tensor. The statistical data of the attitude accuracy errors for different cost function combinations are analyzed in order to understand how external disturbances and inaccurate information about the parameters of the satellite affect the actuated motion.

## 2. Problem Statement

The satellite motion in a circular orbit is considered. It is affected by the gravitational, aerodynamic, and disturbing torques, with the latter having a random nature. The spacecraft should be stabilized in an unstable equilibrium position in the gravitational field.

### 2.1. Equations of Motion

The following coordinate systems are used:

1. $OX_1X_2X_3$ is the inertial frame *J2000* (IF), the axis $OX_3$ is co-directed with the axis of rotation of the Earth, $OX_1$ is directed to the point of the vernal equinox, and the second axis completes the system to the right triple;
2. $OY_1Y_2Y_3$ is the orbital frame (OF), the $OY_3$ axis is directed along the SC radius vector, $OY_2$ coincides with the normal to the satellite orbit, and the third axis $OY_1$ complements the right-hand frame;
3. $Oxyz$ is the reference frame (RF), which describes the required satellite attitude trajectory;
4. $O\xi\eta\varsigma$ is the satellite-fixed frame (SF); its axes coincide with the principal central axes of inertia.

Denote the corresponding direction cosine matrices used in the work as follows:

$$OF \xrightarrow{A} BF, \ OF \xrightarrow{C} RF, \ RF \xrightarrow{D} BF.$$

The direction of the RF axes relative to the OF (Figure 1) is defined by the attitude angles $\alpha$, $\beta$, $\gamma$ (rotation sequence 2-3-1), and matrix **C**:

$$\mathbf{C} = \begin{pmatrix} \cos\alpha\cos\beta & \sin\beta & -\sin\alpha\cos\beta \\ -\cos\alpha\sin\beta\cos\gamma + \sin\alpha\sin\gamma & \cos\beta\cos\gamma & \sin\alpha\sin\beta\cos\gamma + \cos\alpha\sin\gamma \\ \sin\alpha\cos\gamma + \cos\alpha\sin\beta\sin\gamma & -\cos\beta\sin\gamma & -\sin\alpha\sin\beta\sin\gamma + \cos\alpha\cos\gamma \end{pmatrix}. \tag{1}$$

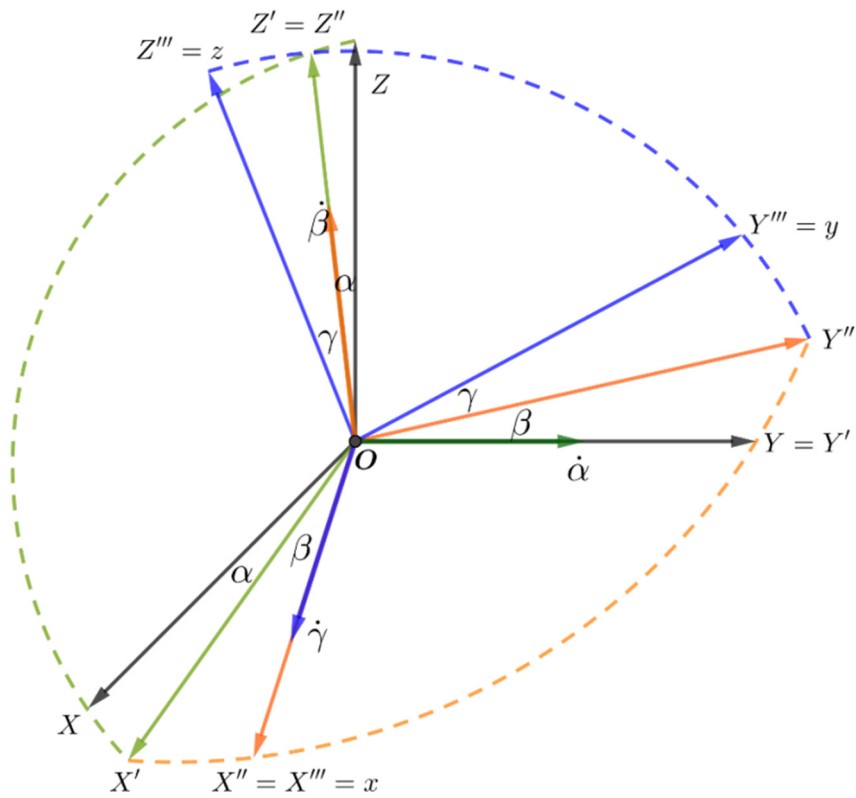

**Figure 1.** Euler's angles (2-3-1). $OXYZ \xrightarrow{OY,\alpha} OX'Y'Z' \xrightarrow{OZ',\beta} OX''Y''Z'' \xrightarrow{OX'',\gamma} OX'''Y'''Z''' \equiv Oxyz$.

This rotation sequence is used because it does not have singularity in the desired mode: $\alpha = 0$, $\beta = 0$, $\gamma = 0$.

The satellite is a rigid body, and its angular motion is described by Euler's equation in the SF and kinematic relations in the form of quaternions, where the quaternion $\mathbf{Q} = (q_0, \mathbf{q})$ and the absolute angular velocity of the spacecraft $\boldsymbol{\omega}_{abs}$ is

$$
\begin{cases}
\mathbf{J}\dot{\boldsymbol{\omega}}_{abs} + \boldsymbol{\omega}_{abs} \times \mathbf{J}\boldsymbol{\omega}_{abs} = \mathbf{M}_{grav} + \mathbf{M}_{aero} + \mathbf{M}_{ctrl} + \mathbf{M}_{dist}, \\
\dot{q}_0 = -\frac{1}{2}\mathbf{q}^T\boldsymbol{\omega}_{abs}, \\
\dot{\mathbf{q}} = \frac{1}{2}(q_0\boldsymbol{\omega}_{abs} + \mathbf{q} \times \boldsymbol{\omega}_{abs}),
\end{cases}
\tag{2}
$$

where $\mathbf{J}$ is the satellite's inertia tensor, $\mathbf{M}_{grav}$ is the gravitational torque, $\mathbf{M}_{aero}$ is the aerodynamic torque, $\mathbf{M}_{ctrl}$ is the control torque, and $\mathbf{M}_{dist}$ is the external disturbing torque. In accordance with [25], the expression for the absolute angular velocity $\boldsymbol{\omega}_{abs}$ is

$$
\boldsymbol{\omega}_{abs} = \mathbf{A}(\boldsymbol{\omega}_0^{OF} + \boldsymbol{\omega}_{ref}^{OF}) + \boldsymbol{\omega}_{rel},
\tag{3}
$$

where $\boldsymbol{\omega}_0^{OF} = (0, \omega_0, 0)^T$ is the orbital angular velocity, $\boldsymbol{\omega}_{ref}^{OF} = \begin{pmatrix} 0 \\ 1 \\ 0 \end{pmatrix}\dot{\alpha} + \begin{pmatrix} \sin\alpha \\ 0 \\ \cos\alpha \end{pmatrix}\dot{\beta} + \begin{pmatrix} \cos\alpha\cos\beta \\ \sin\beta \\ -\sin\alpha\cos\beta \end{pmatrix}\dot{\gamma}$

is the reference angular velocity, $\boldsymbol{\omega}_{rel}$ is the angular velocity relative to the RF, and $\mathbf{A}$ is the direction cosine matrix corresponding to the transition from OF to BF.

The derivative of the absolute angular velocity is

$$
\dot{\boldsymbol{\omega}}_{abs} = \dot{\mathbf{A}}(\boldsymbol{\omega}_0^{OF} + \boldsymbol{\omega}_{ref}^{OF}) + \dot{\boldsymbol{\omega}}_{rel} + \mathbf{A}\dot{\boldsymbol{\omega}}_{ref}^{OF}.
\tag{4}
$$

### 2.2. Models of External Torques

2.2.1. Gravity Gradient Torque

A Newtonian gravitational field is considered. The gravity gradient torque is

$$
\mathbf{M}_{grav} = 3\omega_0^2\mathbf{e}_r \times \mathbf{J}\mathbf{e}_r
\tag{5}
$$

where $\omega_0 = \sqrt{\mu/r^3}$ is the value of the orbital angular velocity, $\mu$ is the gravitational parameter, and $\mathbf{e}_r$ is the unit radius vector direction.

2.2.2. Aerodynamic Torque

The specular–diffuse model is used to describe the aerodynamic effect according to [30]. The shape of the spacecraft is a parallelepiped; that is, the resulting aerodynamic torque is the sum of the torques acting on the spacecraft sides facing the incoming flow. The torque acting on one side is

$$
\mathbf{M}_{aero} = \rho V_0^2\left((1-\varepsilon)\mathbf{J}_1 + 2\varepsilon\mathbf{J}_2 + (1-\varepsilon)\frac{\nu}{V_0}\mathbf{J}_3\right),
\tag{6}
$$

where $\rho$ is the density of the atmosphere, the model parameter $\varepsilon \approx 0.1$ is the fraction of molecules that are reflected specularly, and $\nu/V_0 \approx 0.1$, where $\nu$ is the parameter proportional to the most probable thermal velocity of diffusely reflected molecules, $V_0$ is the value of incoming flow velocity, $\mathbf{e}_{V_0}$ is its unit direction vector, $\mathbf{r}_c$ is the radius vector from the spacecraft's center of mass to the center of the considered side, $\sigma$ is its surface area, and $\mathbf{n}$ is its outer normal unit vector (Figure 2). The expressions for quantities $\mathbf{J}_1 = \left(\mathbf{e}_{V_0}^T\mathbf{n}\right)\mathbf{r}_c \times \mathbf{e}_{V_0}\sigma$, $\mathbf{J}_2 = \left(\mathbf{e}_{V_0}^T\mathbf{n}\right)^2\mathbf{r}_c \times \mathbf{n}\sigma$, and $\mathbf{J}_3 = \left(\mathbf{e}_{V_0}^T\mathbf{n}\right)\mathbf{r}_c \times \mathbf{n}\sigma$ are calculated only when the considered side satisfies the following condition: $\mathbf{e}_{V_0}^T\mathbf{n} > 0$; $\sigma$ is the reference area. It is worth noting that if one calculates the aerodynamic torque for a spherical body using the specular–diffuse model (6), then a more traditional expression will be obtained [31].

$$
\mathbf{M}_{aero} = -\frac{1}{2}\rho V_0^2\sigma c_D\widehat{\mathbf{V}} \times \mathbf{b},
\tag{7}
$$

where $\widehat{\mathbf{V}}$ is the aerodynamic drag acting on the spacecraft's side, $\mathbf{b}$ is the torque arm, and $c_D$ is the coefficient of drag.

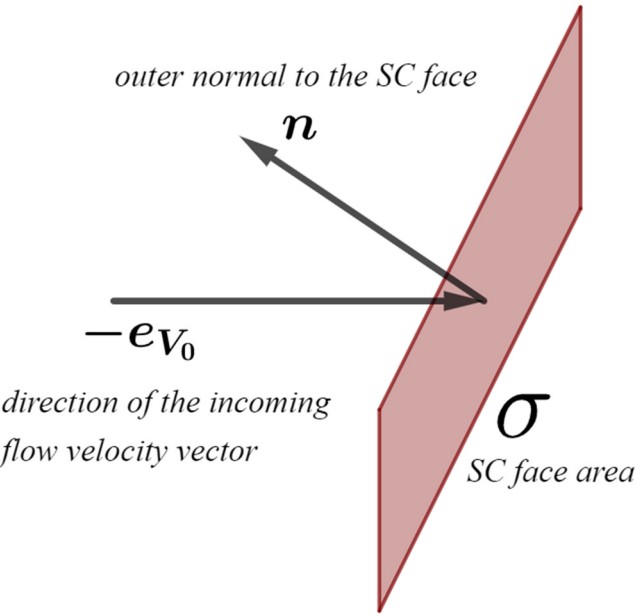

**Figure 2.** Side of the satellite facing the incoming flow.

The center of mass is shifted relative to the geometric center by 1 cm along the second axis of the SF. In this situation, the aerodynamic torque acts as a perturbing one in the required satellite attitude, so corresponding compensation from the MACS is required.

The total aerodynamic torque acting on the spacecraft consists of three (or fewer) torques acting on three (or fewer) spacecraft sides facing the incoming flow. The velocity of the incoming flow is the sum of the spacecraft's velocity and the velocity of the atmosphere, which occurs due to the Earth's rotation

$$\mathbf{V}_0 = \mathbf{V} - \boldsymbol{\omega}_E \times \mathbf{r}, \tag{8}$$

where $\mathbf{r}$, $\mathbf{V}$ are the radius vector and the velocity of the spacecraft, respectively; $\boldsymbol{\omega}_E$ is the angular velocity of the Earth's rotation, $|\boldsymbol{\omega}_E| \approx 7.29 \cdot 10^{-5}\,\text{rad/s}$.

2.2.3. Magnetic Torque and Geomagnetic Field

The magnetic control torque is

$$\mathbf{M}_{magn} = \mathbf{m} \times \mathbf{B} \tag{9}$$

where $\mathbf{m}$ is the spacecraft's dipole moment and $\mathbf{B}$ is the geomagnetic induction vector. This paper considers the direct and inclined dipole models. In the general case, the following expression is used to describe the dipole model:

$$\mathbf{B} = -B_0 \left( \mathbf{k} - 3\left( \frac{\mathbf{k}^T \mathbf{r}}{r} \right) \frac{\mathbf{r}}{r} \right), \tag{10}$$

where $B_0 = \mu_e/r^3$, $\mathbf{k}$ is the unit vector of the dipole axis, and $r = |\mathbf{r}|$.

The dipole unit vector in the direct dipole model in the IF is $\mathbf{k}^{direct} = (0, 0, -1)^T$. The inclined dipole unit vector $\mathbf{k}^{inclined}$ expression is given in [32].

### 2.2.4. Additional External Disturbances

The numerical simulation takes into account the torque of an unknown nature $\mathbf{M}_{dist}$. It includes both a constant perturbing torque and a random one. The value $|\mathbf{M}_{dist}|$ is about 10% of the gravitational torque value.

### 2.3. Control Torque

Equation (2) in the unperturbed case ($\mathbf{M}_{dist} = 0$) is expressed in the form (11). The dynamical equation is expanded taking into account the expression for the absolute angular velocity (4). The kinematic equations are presented in terms of the direction cosine matrices:

$$\begin{cases} \mathbf{J}\dot{\boldsymbol{\omega}}_{rel} = -\boldsymbol{\omega}_{abs} \times \mathbf{J}\boldsymbol{\omega}_{abs} - \mathbf{J}\left(\dot{\mathbf{A}}(\boldsymbol{\omega}_0 + \boldsymbol{\omega}_{ref})\right) - \mathbf{J}\left(\mathbf{A}\dot{\boldsymbol{\omega}}_{ref}\right) + \mathbf{M}_{grav} + \mathbf{M}_{aero} + \mathbf{M}_{ctrl}, \\ \dot{\mathbf{A}} = -[\boldsymbol{\omega}_{abs} - \mathbf{A}\boldsymbol{\omega}_0]_{\times}\mathbf{A}. \end{cases} \tag{11}$$

where $[a]_{\times}$ is the notation of the cross-product matrix, which is composed of the vector $a = (a_1, a_2, a_3)^T$ components as follows:

$$[a]_{\times} = \begin{pmatrix} 0 & -a_3 & a_2 \\ a_3 & 0 & -a_1 \\ -a_2 & a_1 & 0 \end{pmatrix}. \tag{12}$$

To ensure the asymptotic stability, the following Lyapunov function is used to construct a control [33]:

$$V = \tfrac{1}{2}\boldsymbol{\omega}_{rel}^T\mathbf{J}\boldsymbol{\omega}_{rel} + k_a((1 - d_{11}) + (1 - d_{22}) + (1 - d_{33})), \tag{13}$$
$$k_a = const > 0.$$

In this case, the expression for the derivative of the Lyapunov function has the form

$$\begin{aligned} \dot{V} &= \boldsymbol{\omega}_{rel}^T\mathbf{J}\dot{\boldsymbol{\omega}}_{rel} - k_a(\omega_{3,rel}(d_{21} - d_{21}) + \omega_{2,rel}(d_{13} - d_{31}) + \omega_{1,rel}(d_{32} - d_{23})) \\ &= \boldsymbol{\omega}_{rel}^T\mathbf{J}\dot{\boldsymbol{\omega}}_{rel} + k_a\boldsymbol{\omega}_{rel}^T\mathbf{S}_d = \boldsymbol{\omega}_{rel}^T(\mathbf{J}\dot{\boldsymbol{\omega}}_{rel} + k_a\mathbf{S}_d), \end{aligned} \tag{14}$$

where $d_{ij}$ represents elements of matrix $\mathbf{D}$ (RF → BF), $\mathbf{S}_d = (d_{23} - d_{32}, d_{31} - d_{13}, d_{12} - d_{21})^T$, $\boldsymbol{\omega}_{rel} = (\omega_{1,rel}, \omega_{2,rel}, \omega_{3,rel})^T$, and $k_a$ is a scalar positive parameter, $[k_a] = [\text{N} \cdot \text{m}]$.

For the asymptotic stability, by virtue of the Barbashin–Krasovskii–LaSalle theorem [34], it is necessary to ensure the non-positive derivative of the candidate Lyapunov function due to the equations of motion. So, it suffices to keep the equality

$$\dot{V} = \boldsymbol{\omega}_{rel}^T\left(-\boldsymbol{\omega}_{abs} \times \mathbf{J}\boldsymbol{\omega}_{abs} - \mathbf{J}\left(\dot{\mathbf{A}}(\boldsymbol{\omega}_0 + \boldsymbol{\omega}_{ref})\right) - \mathbf{J}\left(\mathbf{A}\dot{\boldsymbol{\omega}}_{ref}\right) + \mathbf{M}_{grav} + \mathbf{M}_{aero} + \mathbf{M}_{ctrl} + k_a\mathbf{S}_d\right) \le 0 \tag{15}$$

Introducing a positive scalar parameter $k_\omega = const > 0$, one rewrites this condition in the form

$$-\boldsymbol{\omega}_{abs} \times \mathbf{J}\boldsymbol{\omega}_{abs} - \mathbf{J}\left(\dot{\mathbf{A}}(\boldsymbol{\omega}_0 + \boldsymbol{\omega}_{ref})\right) - \mathbf{J}\left(\mathbf{A}\dot{\boldsymbol{\omega}}_{ref}\right) + \mathbf{M}_{grav} + \mathbf{M}_{aero} + \mathbf{M}_{ctrl} + k_a\mathbf{S}_d = -k_\omega\boldsymbol{\omega}_{rel}. \tag{16}$$

Taking into account (11) and (16), the control torque has the form

$$\mathbf{M}_{ctrl} = -k_\omega\boldsymbol{\omega}_{rel} - k_a\mathbf{S}_d + \boldsymbol{\omega}_{abs} \times \mathbf{J}\boldsymbol{\omega}_{abs} + \mathbf{J}\dot{\mathbf{A}}(\boldsymbol{\omega}_0 + \boldsymbol{\omega}_{ref}) + \mathbf{J}\mathbf{A}\dot{\boldsymbol{\omega}}_{ref} - \mathbf{M}_{grav} - \mathbf{M}_{aero}. \tag{17}$$

The right side here depends on the reference trajectory parameters and the control gains $k_\omega$ and $k_a$.

Substituting $\boldsymbol{\omega}_{rel} = (0, 0, 0)^T$ and $\mathbf{A} = \mathbf{C}$, $\mathbf{D} = \mathbf{E}$, where $\mathbf{E}$ is the identity matrix, into Equation (17) provides

$$\mathbf{M}_{ctrl} = \boldsymbol{\omega}_{abs} \times \mathbf{J}\boldsymbol{\omega}_{abs} - \mathbf{J}\left[\mathbf{C}\boldsymbol{\omega}_{ref}\right]_{\times}\mathbf{C} + \mathbf{J}\mathbf{C}\dot{\boldsymbol{\omega}}_{ref} - \mathbf{M}_{grav} - \mathbf{M}_{aero}. \tag{18}$$

This control ensures the spacecraft's motion along the reference trajectory in the unperturbed case.

### 2.4. Magnetic Control Torque

It is impossible to produce the control torque component along the geomagnetic induction vector. The torque implemented by the magnetic coils differs from Equation (17) (Figure 3) and is calculated in the SF as follows:

$$\mathbf{M}_{magn} = \mathbf{m} \times \mathbf{AB}_{magn} = \left( \frac{\mathbf{AB}_{magn} \times \mathbf{M}_{ctrl}}{B_{magn}{}^2} \right) \times \mathbf{AB}_{magn} = \mathbf{M}_{ctrl} - \frac{\mathbf{AB}_{magn} \left( \mathbf{AB}_{magn}^{T} \mathbf{M}_{ctrl} \right)}{B_{magn}^2}, \tag{19}$$

where $\mathbf{m} = \mathbf{AB}_{magn} \times \mathbf{M}_{ctrl} / B_{magn}^2$ is the satellite's dipole moment, and $\mathbf{B}_{magn}$ is the geomagnetic induction vector given in the OF.

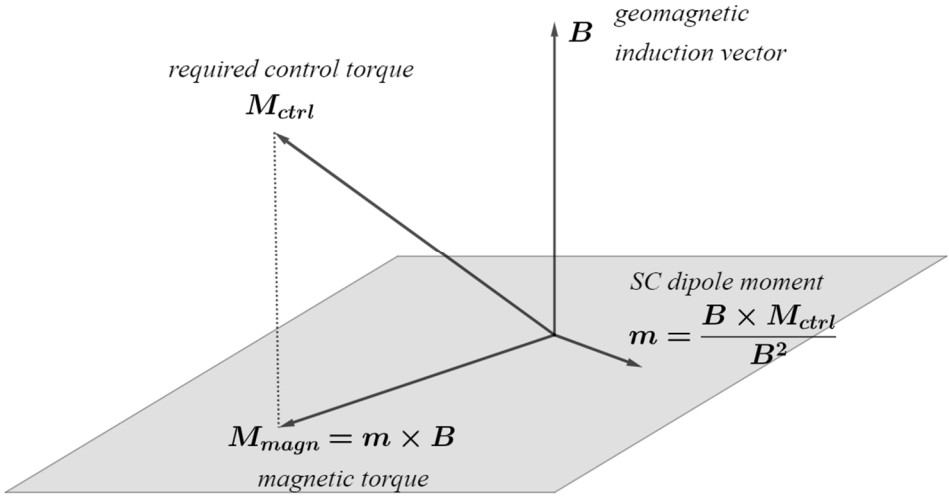

**Figure 3.** Magnetic control torque $\mathbf{M}_{magn}$.

The second term in (19) in the general case is perturbing. It directly affects the attitude accuracy, but it is usually ignored when the control is synthesized. In contrast, as shown in [25], a special reference trajectory in the neighborhood of the required attitude that minimizes this term is constructed. In other words, minimization of the control torque projection onto the geomagnetic induction vector is carried out. The subject of this paper is the reasonable cost function selection for this problem and for the one of the optimal approximation of the geomagnetic field model on a given time interval.

### 3. Reference Trajectory Construction

Below, we briefly describe the PSO method and a two-stage approach proposed in [25]. At the first stage, an algorithm for constructing a special angular trajectory is proposed. The projection of the control torque on the geomagnetic induction vector is minimal on this trajectory. Then, to ensure the asymptotic stability, the control torque (17) with optimal control gains in some sense is constructed. Both stages utilize the global optimization method PSO.

### 3.1. Particle Swarm Optimization Algorithm

PSO is an evolutionary optimization algorithm [28,29] based on a decision-making model of particles' motion in a search for the best solution to the optimization problem. Each particle computes the cost function value for its position in the search space of the problem parameters and receives the information about possible best solutions from its neighbors.

The optimization problem for the swarm is

$$\Phi\left(\mathbf{x}_{p,i}\right) \underset{\mathbb{U}}{\rightarrow} \min, \tag{20}$$

where $\mathbf{x}_{p,i}$ is the position of the particle $p$ at iteration (generation) $i$, which corresponds to the potential best value of the cost function, while $\mathbb{U}$ is the given search area.

At each iteration, the particles chose their motion direction on the next step based on their best position and the best position of the entire swarm (or some neighborhood of the particle)—that is, the best position ever found among all particles (or among the neighbors of the considered particle).

$$\mathbf{x}_{p,i} = \mathbf{x}_{p,i-1} + \mathbf{v}_{p,i},$$

$$\mathbf{v}_{p,i} = c_{in}\mathbf{v}_{p,i-1} + c_{cog}\left(\mathbf{x}_{best,p,i-1} - \mathbf{x}_{p,i-1}\right) + c_{soc}\left(\mathbf{x}_{local\,best,p,i-1} - \mathbf{x}_{p,i-1}\right), \tag{21}$$

where $\mathbf{v}_{p,i}$ is the velocity of the particle $p$ at iteration $i$.

The first term in (21) is the inertial velocity component. It is responsible for the desire of the particle to continue moving in the same direction as in the previous step. The second is the cognitive velocity component, which represents the particle's tendency to return to its best position. The last one is the social velocity component. It shows the overall strive of the swarm or some part of the swarm to move to the best position ever found. The coefficients $c_{in}$, $c_{cog}$, $c_{soc}$ are set randomly in a certain range to vary the contribution of each component.

When all particles gather in the vicinity of the best position of the swarm, and the value of the cost function derivative is small for several iterations in a row, we can assume that the algorithm has found the optimal solution.

### 3.2. Trajectory Parametrization

The first stage of solving the problem is the construction of a reference trajectory. The control torque calculated by (18) should have a minimum projection on the geomagnetic induction vector during the spacecraft's motion along the trajectory. This projection serves as the base for the cost function introduction for the optimization problem. The trajectory is the solution for the problem. However, trajectory parametrization should be established first. Its parameters serve as the particles' coordinates in the PSO routine.

The reference trajectory is constructed in a simplified model of satellite motion. The unknown disturbance torque is zero, the geomagnetic field model is a direct dipole, and the aerodynamic torque (6) is constant in a small vicinity of the equilibrium position. This makes all torques on the right side of the first equation of system (2) periodic; consequently, the reference trajectory is also sought as a periodic function. In [25], the reference trajectory is constructed in the following form:

$$\begin{aligned} \alpha &= a_1 \sin u + a_2 \cos u + a_3 \sin 2u + a_4 \cos 2u, \\ \beta &= b_1 \sin u + b_2 \cos u + b_3 \sin 2u + b_4 \cos 2u, \\ \gamma &= g_1 \sin u + g_2 \cos u + g_3 \sin 2u + g_4 \cos 2u, \end{aligned} \tag{22}$$

where $u = \omega_0 t$ is the argument of latitude, while $a_i$, $b_i$, $g_i$ $(i = 1, 2, 3, 4)$ are 12 parameters of the reference angular trajectory. The angles $\alpha$, $\beta$, $\gamma$ correspond to the direction cosine matrix $\mathbf{C}$. This parameterization uses two eigenfrequencies as a reasonable compromise between the accuracy and the computational complexity. The orbital frequency corresponds to the frequency of the gravitational torque variation. The double orbital frequency is used, since the geomagnetic induction vector in the direct dipole model is a $\pi$-periodic

function [32,35]. The optimal trajectory coefficients $a_i$, $b_i$, $g_i$ $(i = 1, 2, 3, 4)$ are found using the PSO with the cost function of the form

$$\Phi_0^{trajectory} = \frac{1}{N} \sqrt{\sum_{n=0}^{N} \varphi_n^2} \to \min, \quad \varphi_n = \left( \frac{\mathbf{M}_{ctrl}^T \mathbf{AB}_{magn}}{M_{ctrl} B_{magn}} \right)_n, \tag{23}$$

where $\varphi_n$ is the relative value of the control torque projection on the geomagnetic induction vector. It is calculated directly at each time step $n = 0, 1, \dots, N$, where $N$ is the total number of steps.

*3.3. Control Gains Searching*

At this stage, the external disturbances and various model inaccuracies are also not taken into account. Moreover, it is assumed that at the initial moment the SC is already on the reference trajectory found in Section 3.2—that is, $\mathbf{A} = \mathbf{C}$, $\mathbf{D} = \mathbf{E}$, where $\mathbf{E}$ is the identity matrix, and $\boldsymbol{\omega}_{rel} = (0, 0, 0)^T$. Thus, the initial conditions of the spacecraft coincide with the initial parameters of the trajectory—that is, $\alpha(0)$, $\beta(0)$, $\gamma(0)$—and the initial angular velocity $\boldsymbol{\omega}_{abs}(0)$. The trajectory and the velocity are given by the parameters $a_i$, $b_i$, $g_i$ $(i = 1, 2, 3, 4)$ and Expressions (3) and (22), respectively. The search for the optimal gains $k_\omega$ and $k_a$ in the Expression (17) for the control torque calculation is also performed by the PSO.

Here the control torque (17) depends on the phase variables (the state vector), so it cannot be calculated directly. The phase variables are obtained by the numerical integration of the equations of motion (2), while taking into account the control torque (19) implemented by the magnetic system.

The empirically selected cost function at this stage in [25], which is a derivative of the cost function used at the first stage (Formula (17)), is

$$\Phi_0^{gains} = \left( \sum_{n=1}^{N} \left( \left( \boldsymbol{\omega}_{rel}^T \mathbf{AB}_{magn} \right) \cdot \left( \mathbf{M}_{ctrl}^T \mathbf{AB}_{magn} \right) \right)^2 + \sum_{n=1}^{N} \left( \left( \mathbf{S}_d^T \mathbf{AB}_{magn} \right) \cdot \left( \mathbf{M}_{ctrl}^T \mathbf{AB}_{magn} \right) \right)^2 \right) \to \min. \tag{24}$$

Although this cost function gives decent results, there was a lack of theoretical basis. In this paper, the choice of the cost functions is based on the Floquet theory and supported by massive numerical simulation.

## 4. Optimization Problems Statement

The reference trajectory in [25] is constructed in a simplified satellite motion model. In particular, there are no perturbations, and the direct dipole model is used. In this paper, the same motion model is used, but a number of alternative cost functions are proposed to formalize the problem of "minimizing the projection of the control torque on the geomagnetic induction vector". In this case, they are based on the analysis of the linearized model of the angular motion. Additionally, the problem of the optimal approximation of the geomagnetic field for a given time interval is solved.

The Euler's dynamics Equation (11), taking into account (17) and (19), is

$$\mathbf{J}\dot{\boldsymbol{\omega}}_{rel} = -k_\omega \boldsymbol{\omega}_{rel} - k_a \mathbf{S}_d - \frac{\mathbf{B}_{magn} \left( \mathbf{B}_{magn}^T \mathbf{M}_{ctrl} \right)}{B_{magn}^2}. \tag{25}$$

The linearized form of Equation (25) in the vicinity of the constructed reference motion is

$$\dot{\mathbf{y}} = \mathbf{G}(t)\mathbf{y} + f(t), \tag{26}$$

where $\mathbf{y} = (\boldsymbol{\omega}_{rel}, \mathbf{S})^T = \left( (\omega_{rel,1}, \omega_{rel,2}, \omega_{rel,3})^T, (\gamma_{rel}, \alpha_{rel}, \beta_{rel})^T \right)^T$; $\mathbf{G}(t)$ is a $6 \times 6$ matrix, $f(t)$ is a $6 \times 1$ vector, and both are periodic. During the linearization in the vicinity of the reference motion, it is assumed that there are no unaccounted external disturbing torques,

the geomagnetic field model is a direct dipole, and the aerodynamic torque is constant in a small neighborhood of the equilibrium position. This determines the period of $\mathbf{G}(t)$ and $f(t)$, which is $T = \pi/\omega_0$.

The matrix $\mathbf{G}(t)$ of System (26) consists of four $3 \times 3$ blocks

$$\mathbf{G}(t) = \begin{pmatrix} \mathbf{J}^{-1}\mathbf{G}_\omega & \mathbf{J}^{-1}\mathbf{G}_S \\ \mathbf{E}_{3\times 3} & \mathbf{0}_{3\times 3} \end{pmatrix}, \tag{27}$$

where $\mathbf{E}_{3\times 3}$ is the $3 \times 3$ identity matrix, $\mathbf{0}_{3\times 3}$ is a zero matrix, and the $3 \times 3$ matrices $\mathbf{G}_\omega$ and $\mathbf{G}_S$ are obtained by grouping the corresponding terms at $\boldsymbol{\omega}_{rel}$ and $\mathbf{S}$, respectively. Detailed expressions for them are given in [25].

The expression for $f(t)$ includes terms that contain neither $\boldsymbol{\omega}_{rel}$ nor $\mathbf{S}$:

$$f(t) = -\mathbf{Cb}\left(\mathbf{Cb}^T\mathbf{M}^0_{ctrl}\right) = -\frac{\mathbf{CB}_{magn}}{B_{magn}}\left(\frac{\mathbf{CB}^T_{magn}\mathbf{M}^0_{ctrl}}{B_{magn}}\right). \tag{28}$$

This is the control torque component along the geomagnetic induction vector.

The homogeneous system corresponding to (26) is

$$\dot{\mathbf{x}} = \mathbf{G}(t)\mathbf{x}. \tag{29}$$

According to [36], if the eigenvalues of the monodromy matrix (multipliers of System (29)) lie inside the unit circle $\left(|\lambda_i| < 1, \ i = \overline{1,6}\right)$, then System (29) is asymptotically stable. This means that all of its solutions are asymptotically stable. It is also stated in [36] that if a trivial solution $\mathbf{x}_0 \equiv 0$ of a linear homogeneous system (29) is asymptotically stable for $t \to \infty$, then the corresponding linear inhomogeneous System (26) is asymptotically stable. In addition, if all multipliers of the monodromy matrix of (29) differ from unity $(\lambda_i \neq 1)$—which is true for the case of the asymptotically stable systems of differential equations—then (26) has a unique $T$-periodic solution in the following form:

$$\mathbf{y}(t) = \mathbf{X}(t)[\mathbf{E} - \mathbf{X}(T)]^{-1}\left\{\int_0^t \mathbf{X}^{-1}(\tau)f(\tau)d\tau + \mathbf{X}(T)\int_t^T \mathbf{X}^{-1}(\tau)f(\tau)d\tau\right\}, \tag{30}$$

where $\mathbf{X}(t)$ is the fundamental matrix of System (29) normalized at $t = 0$. In the case when System (26) is asymptotically stable, all of its solutions are asymptotically stable. Then, the periodic solution is asymptotically stable too, and all other solutions converge to it. Thus, all solutions converge to Solution (30), which determines the motion in the steady state.

It can be seen that the oscillation amplitude in (30) depends on the inhomogeneous term $f(t)$, the fundamental matrix and, consequently, on the system matrix $\mathbf{G}(t)$. The matrix $\mathbf{G}(t)$ depends on both the reference motion and the control gains, while the inhomogeneous term $f(t)$ depends only on the reference motion. Thus, it is possible to set and solve two independent optimization problems: the first is the minimization of $f(t)$ by selecting the optimal reference motion, and the second is the search for optimal control gains, taking into account the found reference angular trajectory.

### 4.1. Reference Trajectory Optimization Problem

It is necessary to reduce the oscillation amplitude of the resulting periodic solution (30) in order to minimize the tracking errors. So, the first task is the minimization of the value (28). This is the stage of the reference trajectory construction.

Table 1 shows three expressions for the cost functions that formalize this problem, where $\|\mathbf{x}\|_2 := \sqrt{x_1^2 + \ldots + x_n^2}$ is the Euclidian norm, and $\|\mathbf{x}\|_\infty := \max_i |x_i|$ is the infinity norm.

**Table 1.** Expressions for the cost functions in the problem of the reference trajectory construction.

| $\phi_1$ | $\phi_2$ | $\phi_3$ |
|---|---|---|
| $\left\| \frac{\mathbf{CB}_{magn}}{B_{magn}} \left( \frac{\mathbf{CB}_{magn}^T \mathbf{M}_{ctrl}^0}{B_{magn}} \right) \right\|_2$ | $\mathbf{CB}_{magn}^T \mathbf{M}_{ctrl}^0$ | $\left\| \frac{\mathbf{CB}_{magn}}{B_{magn}} \left( \frac{\mathbf{CB}_{magn}^T \mathbf{M}_{ctrl}^0}{B_{magn}} \right) \right\|_\infty$ |

The optimization problems for this stage are

$$\Phi_k^{trajectory} = \max_n |\phi_k|_n \to \min, \ k = 1, 2, 3, \tag{31}$$

where $n = 0, 1, 2, \ldots, N$, $N = T/\Delta t$ is the number of integration steps, $T$ is a period, and $\Delta t$ is a time step. Using the PSO with the cost functions given in Table 1, we can find the optimal coefficients of the reference trajectory in the form (22). Recall that when constructing the trajectory, the direct dipole model is used and there are no external perturbations.

*4.2. Control Gains Optimization Problem*

Ensuring the asymptotic stability is the goal of the Lyapunov control (17). It is achieved by the proper selection of the control gains $k_a$, $k_\omega$. In order to find the optimal coefficients, we also use the PSO and formulate an optimization problem with the following cost function:

$$\Phi^{gains} = \max_i(|\lambda_i|) \to \min, \tag{32}$$

where $\lambda_i$, $i = \overline{1,6}$ are the eigenvalues of the monodromy matrix of System (29). However, only those pairs of control gains $k_a$, $k_\omega$ that lead to the condition $|\lambda_i| < 1$ are suitable. In this case, the trivial Solution (30) will be asymptotically stable.

*4.3. Ideal Case Numerical Simulation*

Reference trajectories are found with each of the cost functions proposed in Table 1 for two heights $h = 550$ km and $h = 650$ km. The density of the atmosphere at these altitudes differs by an order of magnitude. To evaluate the final attitude accuracy in steady state in the OF, which is determined by the amplitude of the periodic Solution (30), the initial conditions of the periodic solution are calculated by the following formula [36]:

$$\mathbf{y}(0) = [\mathbf{E} - \mathbf{X}(T)]^{-1} \mathbf{X}(T) \int_0^T \mathbf{X}^{-1}(\tau) f(\tau) d\tau. \tag{33}$$

Thus, the spacecraft is "placed" directly on the trajectory, brushing aside the process of convergence onto it.

Figure 4 shows three found angular trajectories relative to the OF for each height. The final attitude accuracy in the case of an altitude of $h = 550$ km for all three cost functions is approximately the same and is about $\pm 2°$, and in the case of an altitude of $h = 650$ km the best accuracy is obtained by optimizing with the third cost function $\Phi_3^{trajectory}$ and equals $\pm 0.4°$. Thus, the tracking accuracy of the reference trajectory is worse at higher atmospheric densities.

At this stage, it is impossible to establish which cost function is better for each height, since they all show similar final accuracy. However, their behavior can differ significantly when adding perturbations. The final accuracy is affected by unaccounted perturbations, such as inaccurate information about the satellite's inertia tensor or uncertainty about the density of the Earth's atmosphere. Therefore, before choosing one of the cost functions, it is necessary to analyze the sensitivity of the found reference trajectories to the unaccounted perturbations in each specific case.

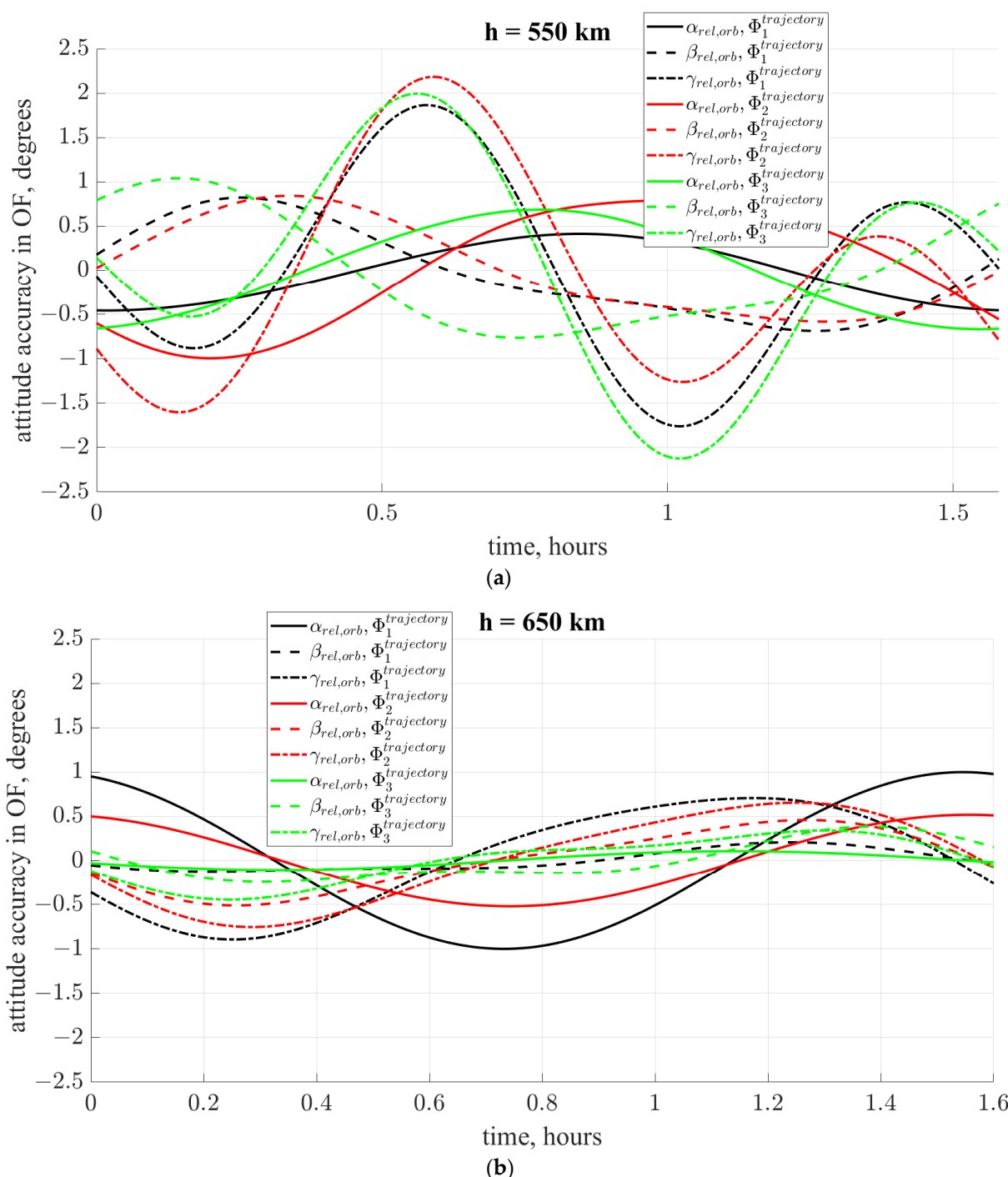

**Figure 4.** Trajectories in OF in steady state for unperturbed problems with different cost functions for different heights: (**a**) $h = 550\,\text{km}$ (**b**) $h = 650\,\text{km}$.

## 5. The Influence of Disturbances

The Lyapunov-based control (17) constructed at the second stage provides the asymptotic stability only for the solutions of the unperturbed Equation (26). Therefore, an accuracy degradation in the full model is expected. This can be estimated using the linearized equations, which in this case have the form

$$\dot{\mathbf{y}} = (\mathbf{G}(t) + \delta\mathbf{G}(t))\mathbf{y} + (f(t) + \delta f(t)), \tag{34}$$

where matrix $\delta\mathbf{G}(t)$ and vector $\delta f(t)$ appear due to the disturbances. These disturbances include the inaccuracy of knowledge of the Earth's atmosphere's density and the spacecraft's inertia tensor, geomagnetic field model $\delta\mathbf{B}$ perturbation, and various other random disturbing torques $\mathbf{M}_{dist}$. It is not possible to take these perturbations into account at the trajectory construction and gains selection stages. Unfortunately, the reference trajectory tracking accuracy usually degrades due to $\delta f(t)$, and in some cases, instability occurs due to $\delta\mathbf{G}(t)$. The second negative effect can be removed by shifting the obtained control gains "deeper" into the stability region. This means choosing the coefficients that are far away from the boundary of the stability region. Trajectory tracking error reduction is possible by choosing a description of the geomagnetic field that is closer to the inclined dipole and using it at the trajectory construction stage. The inclined dipole model, which is used in the numerical simulation, is considered to be the "real" field. The direct dipole model is used during the control construction process. It is considered to be the approximate simplified field representation. The difference between them affects the control performance. However, this effect can be minimized by adjusting the simplified dipole model and using it during the control construction. The deviation of the geomagnetic induction vector in the inclined dipole model $\mathbf{B}_{magn}^{inclined}$ from the model used in the construction steps $\mathbf{B}_{magn}$ is

$$\delta\mathbf{B} = \mathbf{B}_{magn}^{inclined} - \mathbf{B}_{magn}. \tag{35}$$

Consider the expression for the right side of Equation (34), taking into account perturbations only from the inaccuracy of the magnetic field. By analogy with (27), the matrix $\mathbf{G}(t) + \delta\mathbf{G}(t)$ of the perturbed System (34) has a block form and includes terms for the corresponding components of the state vector $\left(\boldsymbol{\omega}_{rel}^T, \mathbf{S}^T\right)^T$. The non-homogeneous part $f(t) + \delta f(t)$ does not contain these quantities; its first term coincides with (28), and the second one is

$$\delta f(t) \approx -2\frac{\mathbf{CB}_{magn}\left(\mathbf{CB}_{magn}^T\mathbf{M}_{ctrl}^0\right)}{B_{magn}^2}\left(\mathbf{B}_{magn}^T\delta\mathbf{B}\right)$$
$$-\frac{\mathbf{C}\delta\mathbf{B}\left(\mathbf{CB}_{magn}^T\mathbf{M}_{ctrl}^0\right)}{B_{magn}^2} - \frac{\mathbf{CB}_{magn}\left(\mathbf{C}\delta\mathbf{B}_{magn}^T\mathbf{M}_{ctrl}^0\right)}{B_{magn}^2}. \tag{36}$$

It depends only on the geomagnetic field model's inaccuracy at the trajectory construction and search for control gains stages. In [25], the direct dipole model is used at these stages—that is, $\mathbf{B}_{magn} = \mathbf{B}_{magn}^{direct}$ in (35).

It is not possible to ensure $\delta\mathbf{B} = 0$. The inclined dipole model $\mathbf{B}_{magn}^{inclined}(t)$ cannot be used for the reference trajectory construction, since it is necessary to use only periodic field models. The direct dipole model is a natural choice for this purpose. However, we can minimize $\delta f(t)$ (36) if a new description of the magnetic field $\mathbf{B}_{magn} = \mathbf{B}_{magn}^{oblique}$ is used. It must simultaneously satisfy two requirements:

1.  $\mathbf{B}_{magn}^{oblique}(t)$ is a periodic function, with the same period as the direct dipole model;
2.  The difference between $\mathbf{B}_{magn}^{oblique}(t)$ and $\mathbf{B}_{magn}^{inclined}(t)$ in a given time interval is in some sense less than the difference between $\mathbf{B}_{magn}^{direct}(t)$ and $\mathbf{B}_{magn}^{inclined}(t)$.

The index "oblique" means the new (desired) dipole model. Essentially, the oblique dipole is tilted relative to the Earth's rotational axis but, in contrast to the inclined dipole, its attitude is fixed in the inertial space. This dipole may provide better approximation than the direct one by being closer to the "real" inclined dipole (Figure 5). To describe the geomagnetic induction vector in the general dipole model, Expression (10) is used, which for the oblique dipole model takes the following form:

$$\mathbf{B}_{magn}^{oblique} = -B_0\left(\mathbf{k}^{oblique} - 3\left(\frac{\left(\mathbf{k}^{oblique}\right)^T\mathbf{r}}{r}\right)\frac{\mathbf{r}}{r}\right), \tag{37}$$

where $\mathbf{k}^{oblique} = (\cos\varphi\sin\theta, \ \sin\varphi\sin\theta, \ -\cos\theta)^T$ is the unit vector of the oblique dipole axis, specified using its attitude angles $(\varphi, \ \theta)$ relative to the IF.

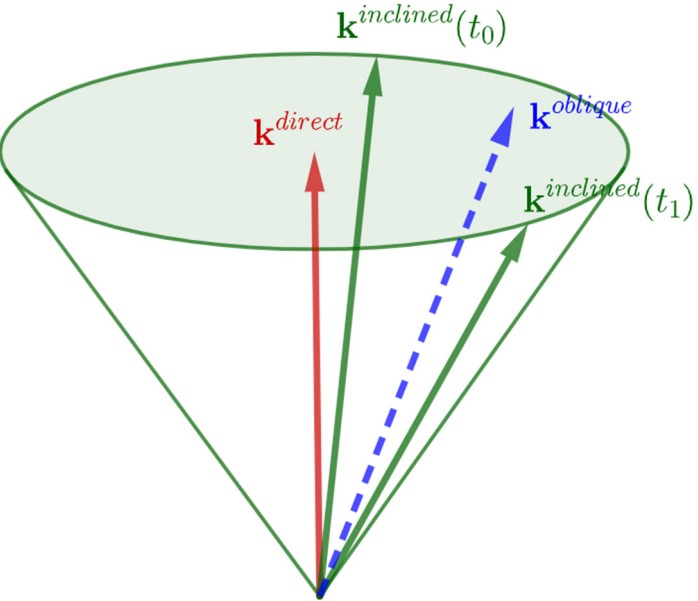

**Figure 5.** Unit vectors of the direct, inclined, and oblique dipole axes. Vectors $\mathbf{k}^{direct}$ and $\mathbf{k}^{oblique}$ do not change over time, and the vector $\mathbf{k}^{inclined}$ rotates.

The optimal values of these angles are found using the PSO with a cost function of the following form:

$$\Phi_k^{dipole} = \max_n (\delta\beta_k)_n \to \min, \ k = 1, 2, 3, 4, \tag{38}$$

where the expressions for $\delta\beta_k$ are given in Table 2, $n = 0, 1, \ldots, N_B$, and $N_B$ is the averaging interval expressed in the number of iterations of the numerical simulation of the equations of motion.

**Table 2.** Expressions for the cost functions in the problem of approximating the magnetic field.

| $\delta\beta_1$ | $\delta\beta_2$ | $\delta\beta_3$ | $\delta\beta_4$ |
|---|---|---|---|
| $\dfrac{\|\mathbf{B}_{magn}^{inclined} - \mathbf{B}_{magn}^{oblique}\|_2}{\|\mathbf{B}_{magn}^{inclined}\|_2}$ | $\dfrac{\|\mathbf{B}_{magn}^{inclined} - \mathbf{B}_{magn}^{oblique}\|_\infty}{\|\mathbf{B}_{magn}^{inclined}\|_\infty}$ | $\|\mathbf{B}_{magn}^{inclined} - \mathbf{B}_{magn}^{oblique}\|_2$ | $\dfrac{\|\mathbf{B}_{magn}^{inclined} - \mathbf{B}_{magn}^{oblique}\|_\infty}{\|\mathbf{B}_{magn}^{inclined}\|_2}$ |

A correctly chosen interval $N_B$ improves the approximation of the magnetic field compared to the direct dipole. The reference trajectory is constructed for one orbital revolution. Therefore, the minimum possible interval for the optimal approximation of the geomagnetic field is also equal to one revolution. That is, it is best to approximate the field for each consecutive orbit. This requires the trajectory coefficients (with a new field) and control gains (with a new trajectory) to be recalculated for each orbit (Figure 6). This comes at significant computational costs of optimization problems, since all three optimization problems must be solved at each time interval to improve the final attitude accuracy. In addition, frequent changing of the reference trajectory leads to the inevitable transient processes. Moreover, the settling time allocated for each reference trajectory may not even be enough to achieve it before switching to the new reference trajectory. Therefore, the averaging interval should be more than one orbit and less than a 24 h interval, because the direct dipole model is an averaging of the inclined dipole over 24 h.

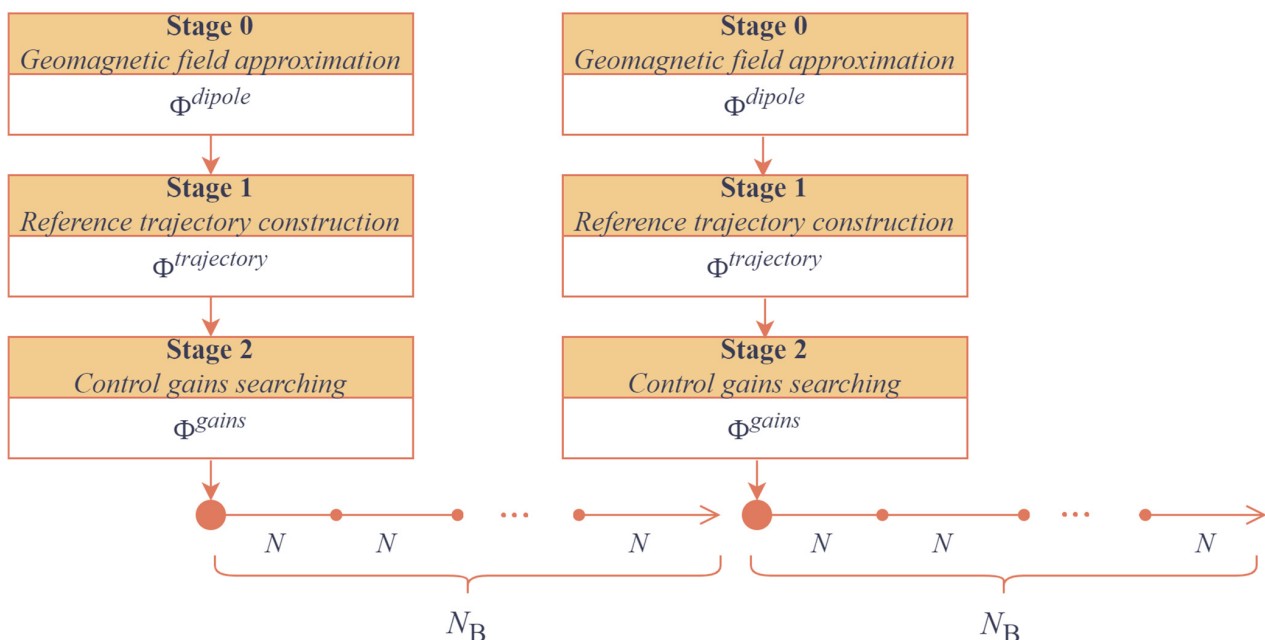

**Figure 6.** General scheme of the trajectory construction algorithm.

This paper considers the averaging intervals equal to six orbital revolutions $N_B = 6N$. The trajectories are constructed using the PSO for different sets of cost functions given in Table 3. The cost function set № 1.2, for example, means that the cost function $\Phi_1^{trajectory}$ is chosen to construct the reference trajectory and the cost function $\Phi_2^{dipole}$ is chosen for the approximation of the geomagnetic field dipole model. The cost function for the control gains optimization problem in all sets is the same $\Phi^{gains}$ (32).

**Table 3.** Sets of optimization problems for the reference trajectory construction.

|  | $\Phi_1^{trajectory}$ | $\Phi_2^{trajectory}$ | $\Phi_3^{trajectory}$ |
|---|---|---|---|
| Direct dipole | № 1.0 | № 2.0 | № 3.0 |
| $\Phi_1^{dipole}$ | № 1.1 | № 2.1 | № 3.1 |
| $\Phi_2^{dipole}$ | № 1.2 | № 2.2 | № 3.2 |
| $\Phi_3^{dipole}$ | № 1.3 | № 2.3 | № 3.3 |
| $\Phi_4^{dipole}$ | № 1.4 | № 2.4 | № 3.4 |

Figure 7 shows the resulting reference trajectories for each pair of optimization problems for an altitude of 650 km. Each curve shows the change in the worst tracking angle of the reference trajectory over a six-orbit interval for each optimization case. The perturbation in these examples is only the geomagnetic field model perturbation at the stage of the trajectory construction.

Figure 7 shows that it is not obvious which set of cost functions is the best, especially when numerically simulated in the full model. It is necessary to analyze the system behavior under various perturbations.

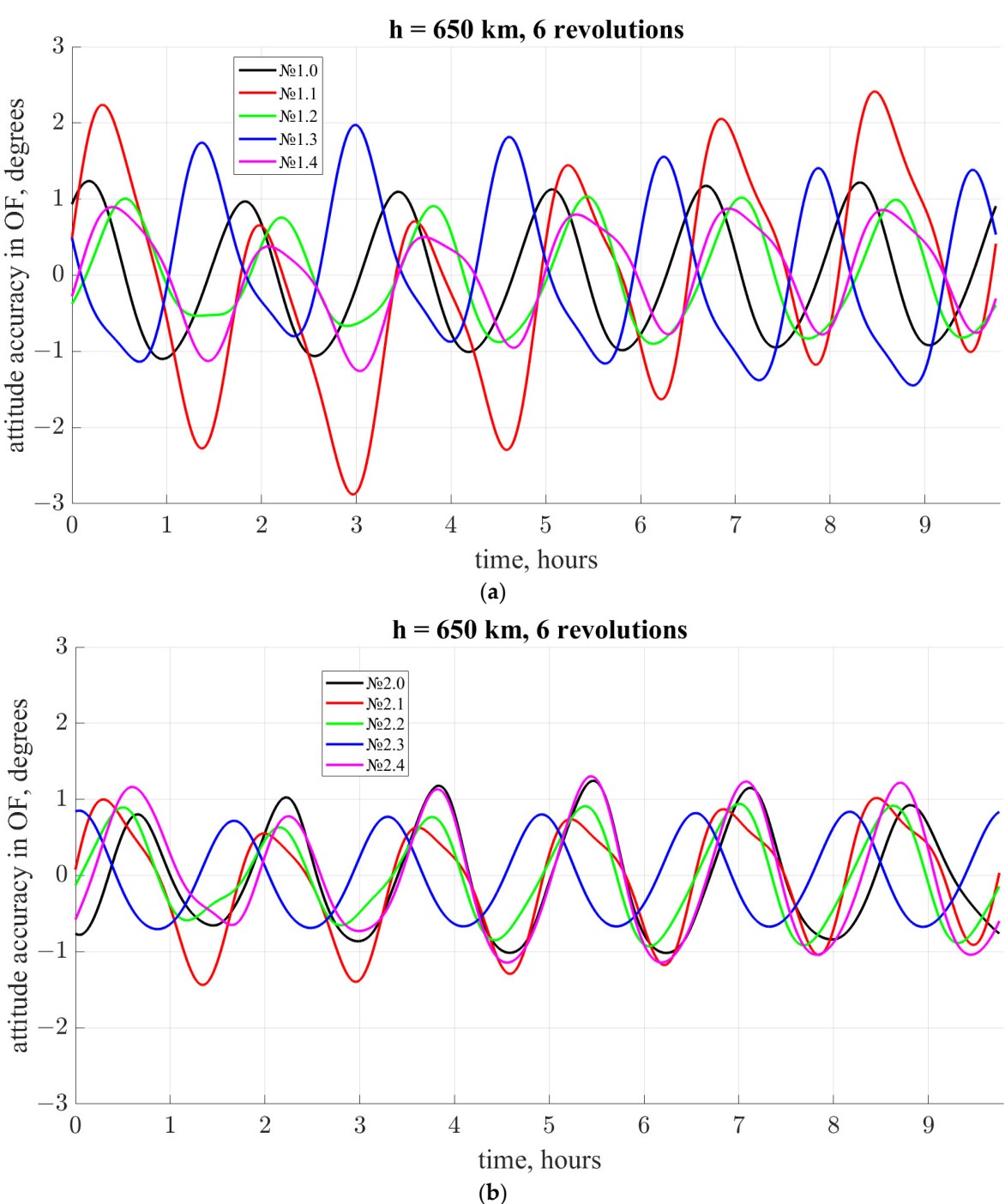

**Figure 7.** *Cont.*

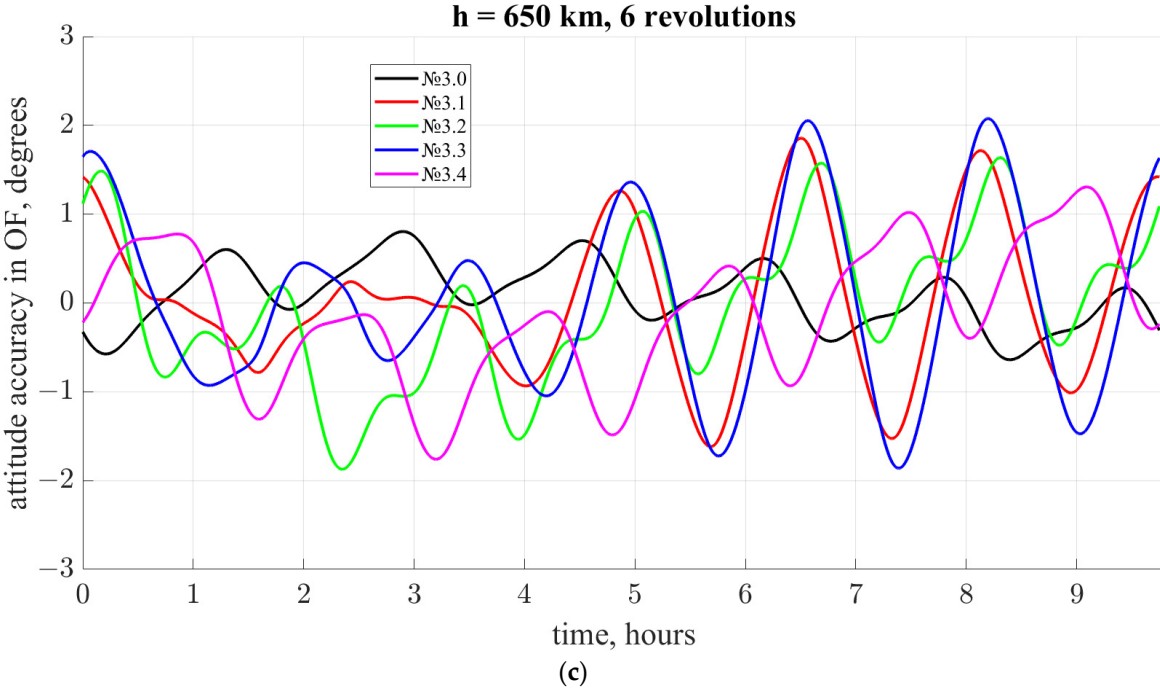

(c)

**Figure 7.** Angular trajectories (in the OF) in the inclined dipole models for various cost functions at an altitude of 650 km, 6-orbit magnetic field approximation: (**a**) set of optimization problems for the reference trajectory construction with $\Phi_1^{trajectory}$ and different dipole models; (**b**) set of optimization problems for the reference trajectory construction with $\Phi_2^{trajectory}$ and different dipole models; (**c**) set of optimization problems for the reference trajectory construction with $\Phi_3^{trajectory}$ and different dipole models.

The inaccurate knowledge of the inertia tensor has the most significant role in the tracking errors of the reference trajectory and can even lead to instability. To compare the tracking accuracy of the reference trajectories in the steady state in numerical simulation, we consider different perturbations of the inertia tensor:

$$\delta \mathbf{J} = diag(\delta J_1, \delta J_2, \delta J_3),\tag{39}$$

where $\delta J_i \in [0.95, \ 1.05], \ i = \overline{1,3}$. Thus, the perturbed inertia tensor is

$$\mathbf{J}_{dist} = \mathbf{J}\delta\mathbf{J},\tag{40}$$

where $\mathbf{J} = diag(J_1, J_2, J_3)$ is the unperturbed spacecraft inertia tensor.

Almost 10,000 simulation runs were performed. Figure 8 shows the distribution of the worst values of the reference trajectory tracking accuracy—that is, the maximum relative attitude deviation in all angles $\alpha_{rel}$, $\beta_{rel}$, $\gamma_{rel}$ for each $\mathbf{J}_{dist}$ for the orbit height 650 km. The inaccurate knowledge of the inertia tensor greatly degrades the attitude accuracy fairly often. However, the sets of cost functions № 1.2 and № 1.4 demonstrate acceptable results. The final attitude accuracy remains about 3–5 degrees for any perturbations of the inertia tensor within 5% of the nominal values of the inertia moments.

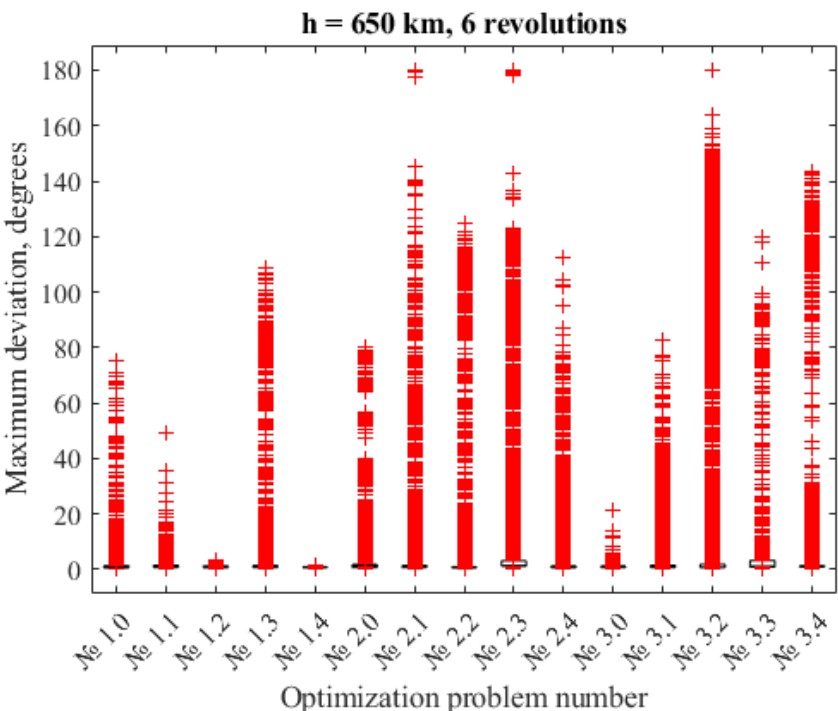

**Figure 8.** Distribution of the worst values of reference trajectory tracking accuracy for orbit height 650 km.

## 6. Full Model Numerical Simulation

As the numerical example, a reference trajectory for three intervals of six orbital revolutions each is constructed, so the total simulation time is $T = 18T_0 \approx 30\,\text{h}$. Each interval has its own oblique dipole axis $\mathbf{k}^{oblique}$. The cost function set № 1.4 is used. Figure 9 shows the result obtained using the described three-stage approach (Figure 6). The attitude accuracy in this case is 1.5 degrees.

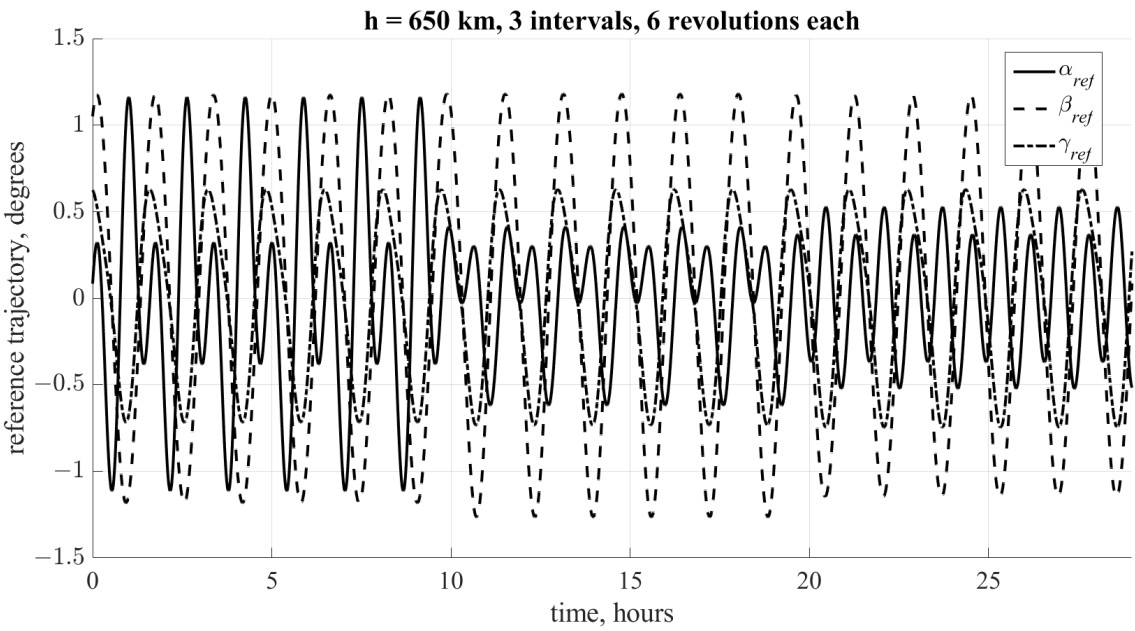

**Figure 9.** Reference trajectory obtained with 3 intervals with different oblique dipole geomagnetic field approximation.

The numerical simulation was carried out with the following initial conditions and disturbances (Table 4) and is shown in Figure 10. The final attitude accuracy in a steady state is about $\pm 2°$. Figure 11 shows the relative angular velocity $\boldsymbol{\omega}_{rel}$. Figure 12 shows the satellite dipole moment **m**. Thus, the correct selection of the cost functions makes it possible to achieve good accuracy both in the presence of external unaccounted disturbances, and with inaccurate knowledge of the motion models.

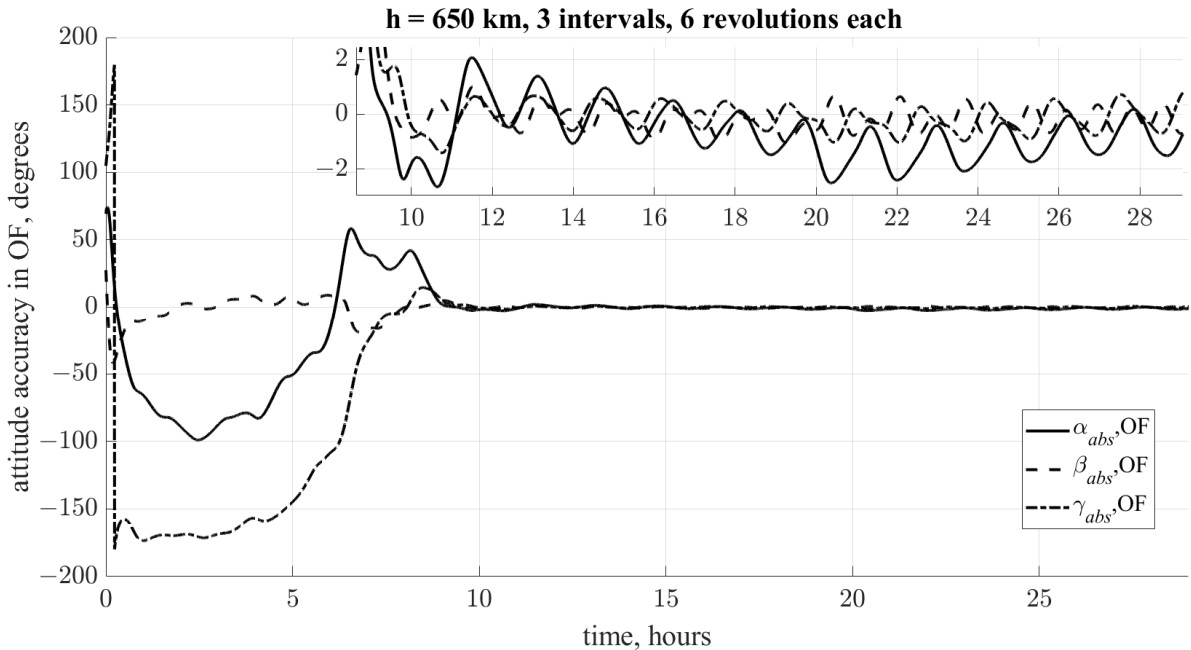

**Figure 10.** Final attitude accuracy in the orbital frame.

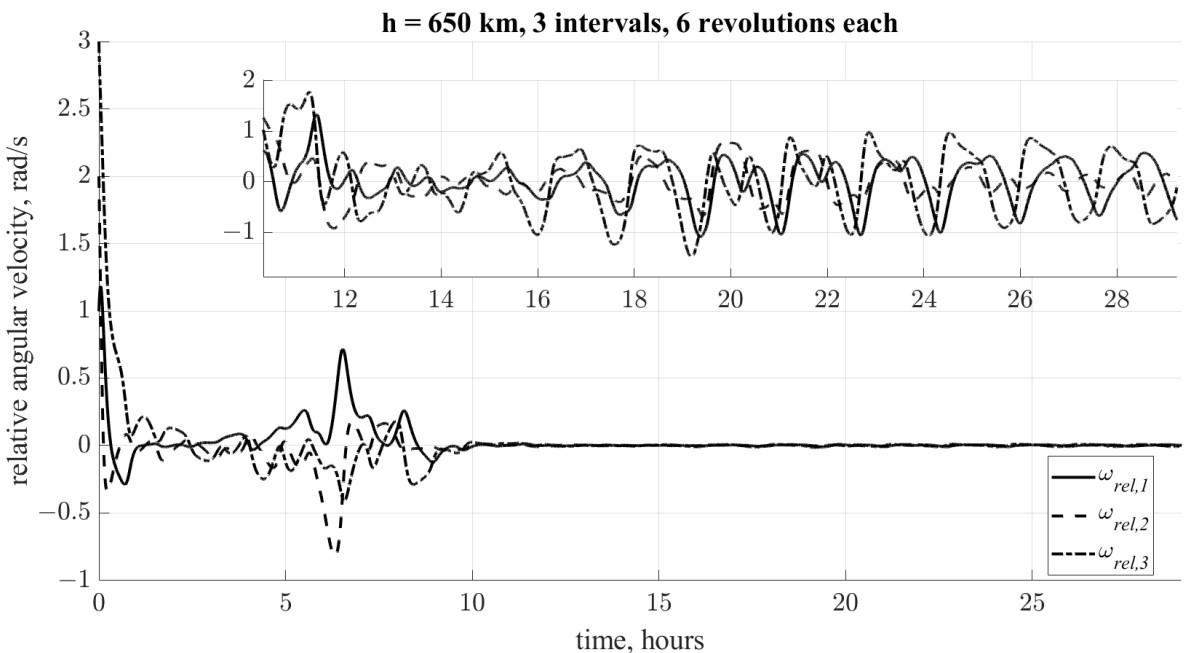

**Figure 11.** Relative angular velocity.

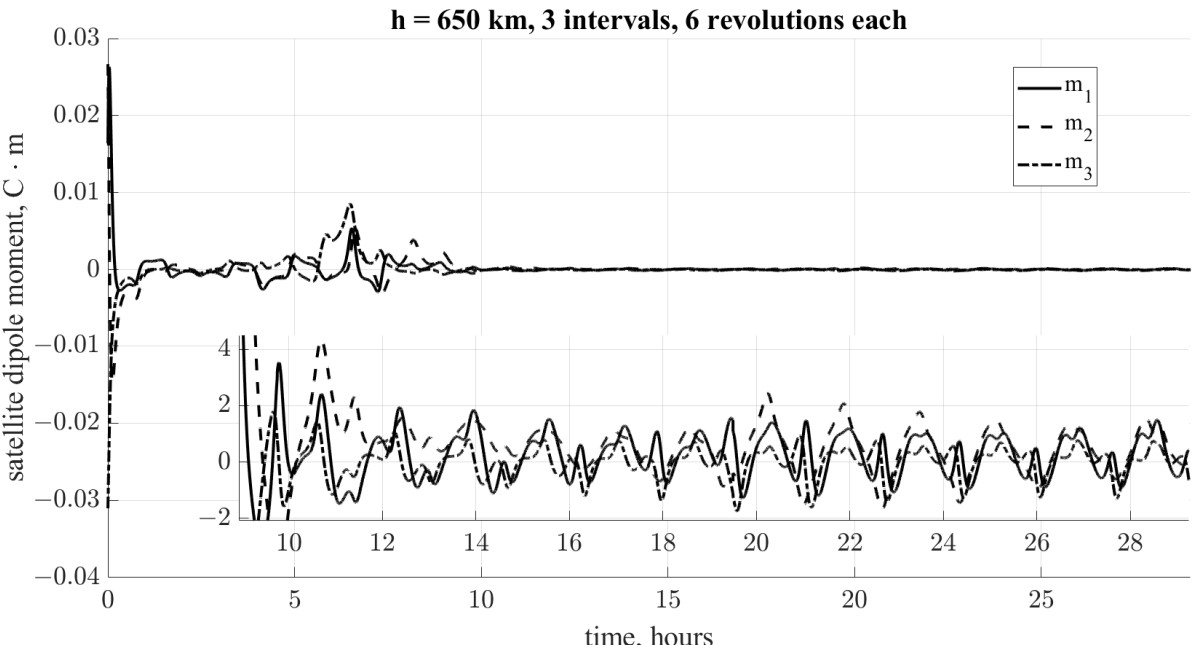

**Figure 12.** Satellite dipole moment.

**Table 4.** Parameters for the numerical simulation.

| Name | Value |
|---|---|
| Simulation time | $T = 18T_0 \approx 30\,\text{h},$ |
| SC initial angular velocity | $\boldsymbol{\omega}_{abs} = (1, 2, 3)^T \cdot 10^{-3}$ rad/s |
| SC initial attitude | $\alpha_{rel} = 55°$ |
| | $\beta_{rel} = 55°$ |
| | $\gamma_{rel} = 55°$ |
| Magnetic field model | IGRF |
| Inaccuracy of knowledge of the density of the atmosphere | 20% |
| Inaccuracy of knowledge of the SC inertia tensor | 5% |
| External random disturbances | $|\mathbf{M}_{dist}| = 10^{-9}$ N · m |

## 7. Conclusions

The paper analyzes the factors that affect the attitude accuracy when solving the problem of a three-axis magnetically controlled motion of a spacecraft. In addition to the obvious impact of external unaccounted perturbations on the final accuracy, it is shown that in some cases the "unconscious" choice of the cost functions can lead to a strong deterioration in accuracy. Correct problem statement and formalization is an important part of any problem's solution, and this can greatly affect the result. Numerical analysis of the selected cost functions, together with the analytical prerequisites for its selection, should help to improve the result. In this work, two sets of cost functions are identified, which provide good accuracy in the steady state (about 2 degrees), despite all of the perturbations and inaccuracies in the knowledge of the models and the parameters of the SC.

**Author Contributions:** Conceptualization, D.R. and S.T.; investigation, S.T. and A.O.; methodology, S.T. and A.O.; verification, S.T.; results analysis, S.T. and A.O.; formal analysis, D.R.; software implementation, A.O.; writing—original draft preparation, A.O.; writing—review and editing, D.R. and S.T. All authors have read and agreed to the published version of the manuscript.

**Funding:** This research received no external funding.

**Data Availability Statement:** Not applicable.

**Conflicts of Interest:** The authors declare no conflict of interest.

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
