# Peer review of "Comparative Cost Functions Analysis in the Construction of a Reference Angular Motion Implemented by Magnetorquers"

_aerospace, doi:10.3390/aerospace10050468_

Round 1

Reviewer 1 Report

The content of the paper is interesting and well presented. I only have a few minor comments reported in the attached file.

Author Response

We are grateful to the reviewers for their valuable remarks. We hope that our responses will make the paper better.

  1. Links to articles added to the line 33.
  2. At the beginning of the sentence, "The satellite is a rigid body" is added (line 121).
  3. The explanation about the model used and its parameters is added (lines 143-160). If we calculate the torque for a spherical body using the specular-diffuse model, then we get exactly more traditional form (the formula that you mentioned).
  4. The problem appear only after conversion from .docx to .pdf. Original figures in .docx files seem to have good resolution. We uploaded the .png source files to the system, and we hope that the editors will use the originals in order to preserve the quality of the figures in the final version of the article.
  5. Simulation results using the IGRF model are added in Section 6.

Reviewer 2 Report

The paper considers a construction procedure of a satellite reference angular motion in the vicinity of an unstable gravitational equilibrium position. However, there are some problems need to be modified.

1) Why is the Euler angle of the reference frame relative to the orbital system ordered by 2-3-1?

2) The meaning of the Matrix A in formula (3) is not given.

3) Why is there no fork multiplier term in Equation 9?

4) What is the physical meaning of the optimization objective representation given by Equation (18)? How to reflect the optimization of control parameters?

5) What is the relationship between parts 3 and 4 of the article? Why both parts establish their own optimization goals is not clearly explained in the article

6) Section 4.3 and Section 6 have the same section name, which can be distinguished. In addition, the final simulation results of the sixth section have less content, and it is suggested that simulation results of other variables can be given, such as angular velocity, control torque, etc.

7) Compared to Reference 23, what is the innovation of this article? It needs to be given explicitly in the introduction

Author Response

We are grateful to the reviewers for their valuable remarks. We hope that our responses will make the paper better.

  1. In some English-language sources, it is customary to call any sequence of rotations “Euler angles”, but somewhere different sequences have their own names. We decided to remove the mention of “Euler angles” to avoid confusion and simply call them attitude angles. This sequence is used because it does not have singularity in the desired mode (0,0,0). Added clarification to the article, lines 114 and 117.
  2. Added to the lines 132-133.
  3. Mistype in formula (9) is fixed.
  4. This cost function was used in the article (Okhitina A., Roldugin D., Tkachev S. Application of the PSO for the construction of a 3-axis stable magnetically actuated satellite angular motion // Acta Astronaut. Pergamon, 2022. Vol. 195. P. 86–97.). It is a derivative of cost function used at the first stage (formula (17)) with respect to  and . It shows better performance on the second stage (explanation is added to lines 311-316). But the present paper uses more understandable cost function, rather than empirically selected one.
  5. and 7.  It Section 3 we briefly describe the PSO method and a two-stage approach proposed in (Okhitina A., Roldugin D., Tkachev S. Application of the PSO for the construction of a 3-axis stable magnetically actuated satellite angular motion // Acta Astronaut. 2022. Vol. 195. P. 86–97.). In that the cost functions were chosen empirically. In Section 4 we improve this approach and describe in detail how to better choose cost function for a given problem. In this article there is a justification for the choice of cost functions using the Floquet theory. This eventually allows getting better and more reliable results. In the introduction the explanation is added (lines 78-81).
  6. We have renamed the 4.3 and 6 for clarity. Simulation results of other variables are added in Section 6. In addition, the IGRF model was used.